# Early OCTA Changes of Type 3 Macular Neovascularization Following Brolucizumab Intravitreal Injections

**DOI:** 10.3390/medicina58091180

**Published:** 2022-08-30

**Authors:** Anthony Gigon, Maria Vadalà, Vincenza M. E. Bonfiglio, Michele Reibaldi, Chiara M. Eandi

**Affiliations:** 1Department of Ophthalmology, University of Lausanne, Jules-Gonin Eye Hospital, Fondation Asile des Aveugles, 1004 Lausanne, Switzerland; 2Biomedicine, Neuroscience and Advance Diagnostic (BIND) Department, University of Palermo, 90133 Palermo, Italy; 3Department of Surgical Sciences, University of Torino, 10122 Turin, Italy

**Keywords:** neovascular age-related macular degeneration, type 3 neovascularization, retinal angiomatous proliferation, brolucizumab, optical coherence tomography angiography, intravitreal injection

## Abstract

Background and Objectives: Brolucizumab is a novel anti-vascular endothelial growth factor (VEGF), whose efficacy has been shown in the Hawk and Harrier phase 3 clinical studies. The goal of the present case series is to report initial results of brolucizumab intravitreal injections (IVI) on type 3 neovascularization in neovascular age-related macular degeneration (nAMD), evaluated by optical coherence tomography angiography (OCTA). Materials and Methods: This is a bicentric retrospective case series. Patients with newly diagnosed type 3 MNV treated with brolucizumab IVI and at least 6 months follow-up were enrolled. OCTA en face images and B-scans were analyzed for lesions at baseline, 1 month, 3 months, and 6 months. Whenever detectable, lesion area on outer retina and choriocapillaris layers was measured. Results: Twelve eyes of 12 patients were included into the study. The most consistent OCTA sign at baseline was the presence of a vascular tuft in the outer retina (100%). The highest response was achieved at 3 months, with statistically significant decrease in lesion detection in the outer retina, in the choriocapillaris, and outer retinal lesion size. At 6 months, 58% of outer retinal lesions had disappeared. Conclusions: Brolucizumab IVI shows a good short-term efficacy for the treatment of type 3 neovascularizations. Further studies with greater number of patients and longer follow-up are warranted to confirm these findings.

## 1. Introduction

Neovascular age-related macular degeneration (nAMD) can be classified according to the location of the macular neovascularization (MNV). Donald Gass was the first to use the terms type 1 and type 2 neovascularization, corresponding to lesions developing beneath and above the retinal pigment epithelium (RPE), respectively [1]. Early type 3 MNV corresponds to an intraretinal neovascularization, also known as retinal angiomatous proliferation [2], which can, in turn, progress towards the formation of a retino-choroidal anastomosis and eventually a pigment epithelium detachment (PED) in the late phase of the disease [3]. While the location and origin of type 1 and 2 MNV have been established with little debate, the origins of type 3 lesions have not always been clear [2,4]. Currently, sufficient evidence exists to support an origin from the retina, with further downward extension towards the choroid as the lesion progresses [5]. Type 3 MNV is the second most common MNV type, representing 34% of cases in a study by Jung et al. [6], behind type 1 MNV (40%), while pure type 2 MNV is the rarest (9%) [6].

Although differences exist in terms of prognosis of each MNV type, the first-line treatment of nAMD as a whole consists of intravitreal injections (IVI) of anti-vascular endothelial growth factor (VEGF) drugs. Brolucizumab is a novel anti-VEGF first approved in the USA in 2019. Its efficacy in the management of nAMD has been demonstrated in the phase 3 studies of Hawk and Harrier [7,8].

While these phase 3 studies showed the overall good efficacy of Brolucizumab in nAMD, subgroups according to MNV type were not evaluated. Given the anatomical differences of MNV types, nuances in the treatment response could be expected. The goal of the present case series is to report initial results of Brolucizumab IVI on early type 3 MNV in nAMD, evaluated by optical coherence tomography angiography (OCTA).

## 2. Materials and Methods

This is a bicentric retrospective case series. The study protocol and analysis adhered to the tenets of the Declaration of Helsinki and the research protocol was approved by the local Ethical Committees (CERVD n. 2017-00493 (approval on 1 June 2017) and AOUP n. 3/2022 (approval on 13 July 2021)). We reviewed the charts of patients with newly diagnosed nAMD and type 3 MNV treated with intravitreal injections of brolucizumab at the medical retina department of the Jules Gonin Eye Hospital, Lausanne, Switzerland and of the Giaccone University Hospital, Palermo, Italy. Main inclusion criteria were naive nAMD and active early stage type 3 MNV, diagnosed by optical coherence tomography (OCT), fluorescein (FA), and indocyanine green angiography (ICGA). In particular, the following were considered characteristics of type 3 lesion: the presence of cystoid macular edema associated with intraretinal hyperreflective foci (HRF), RPE disruption, and/or a “kissing sign” between the RPE and the neurosensory retina on OCT B-scans. The presence of an intraretinal neovascular net and a hyperfluorescent “hotspot” on late-stage FA and ICGA confirmed the diagnosis. Patients received three monthly injections of brolucizumab in the loading phase. Re-treatment in the maintenance phase was administered at two or three months interval at discretion of the investigators when a recurrence of intra- or subretinal fluid was present on OCT B-scans and/or a new flow signal was evident at the level of the outer retina or choriocapillaris on OCTA. Flow signal changes on OCTA alone without recurrence of fluid on OCT B-scan was not a re-treatment criterion. A follow-up period of at least 6 months after the first IVI of Brolucizumab was required for enrollment into the study.

Patients with a type 1 or 2 lesion, with any other confounding retinal disease or with poor imaging documentation were excluded. Demographic data such as age and sex were retrieved.

Best-corrected visual acuity (BCVA) was measured with the early treatment diabetic retinopathy study (ETDRS) charts at baseline, at 1- 3- and 6 months. The Logarithm of Minimal Angle of Resolution (logMAR) system was used for statistical analysis. The occurrence of adverse events was also investigated, including intraocular inflammation of the anterior chamber, vitreous, or the retina, and occlusive vasculitis.

B-scans and OCTA images were acquired using the RTVue XR Avanti OCTA system (AngioVue system, Optovue^®^ Inc., Fremont, CA, USA) at baseline and at follow-up periods of 1, 3, and 6 months after treatment initiation. The OCTA scanning patterns were 6 × 6 mm and 3 × 3 mm volume scans centered on the fovea. OCTA images analysis was performed by a retina specialist (AG) and a senior ophthalmic photographer. In case of discrepancy, images were discussed with a senior retina specialist (CE) until agreement was reached.

Qualitative features of the MNV on OCTA were analyzed on the en face images at different layers and on the B-scans. In particular, within the deep capillary plexus, we evaluated the presence of dilated vessels. Slabs through the outer retina and through the choriocapillaris were used to detect neovascular tufts. On the B-scans, we analyzed the presence of flow signal associated with the presence of RPE effraction, defined as anastomotic flow passing from the retina through an opened retinal pigment epithelium, and/or HRF, and the presence of an anastomosis between the retinal and choroidal vascular system (downward flow). Segmentations were manually performed when the automatic inbuilt segmentation software was inadequate. Careful attention was given to exclude projection artifacts when an abnormal signal was present. Inbuilt follow-up mode was used during acquisition and helped to ensure the studied locations were identical throughout the follow-up period, even when the lesion was not visible anymore.

Quantitative analysis consisted of MNV flow measurements using the inbuilt semiautomatic algorithm commercially available on the OCT-A device. By manually selecting the area around the lesion, an area of flow signal is computed by the machine and displayed in mm^2^. These measurements were performed in the outer retina and choriocapillaris slabs. When no vascular tuft was visible, we assumed an area of 0 mm^2^.

Qualitative analysis of the B-scan OCT was also performed in terms of presence/absence of intra or subretinal fluid or pigment epithelium detachment. The central retinal thickness (CRT) was measured in the 1 mm central ETDRS grid using the inbuilt automatic tool.

Data were collected and statistics were performed in IBM SPSS Statistics for Windows version 27.0.1 (IBM, New York, NY, USA). Wilcoxon signed rank tests for paired data were used to compare continuous values with non-normal distributions, such as lesion area. Comparisons of proportions of binary variables, such as presence/absence of an OCTA signs, were performed with McNemar’s exact tests. *p*-values < 0.05 were considered statistically significant.

## 3. Results

In total, 12 eyes of 12 patients were included. Mean age at diagnosis was 80.3 ± 7.0 (range 68–94) years. Six (50%) patients were male. One (8%) subject had diabetes type 2 for 10 years, 5 (42%) and 3 (25%) patients received treatment for blood hypertension and hypercholesterolemia, respectively. Six (50%) eyes were pseudophakic with cataract surgery performed more than 5 years before. Mean baseline BCVA was 20/30 (range 20/60–25/20). No systemic or intraocular adverse events were documented throughout the follow-up period. The total number of IVI at 6 months was on average 3.33 (range 3–4). Only three out of 12 eyes received one IVI in the maintenance phase after the first three monthly treatments. All three eyes presented new flow signal on outer retina or choriocapillaris and recurrent intra or subretinal fluid.

A summary of OCTA characteristics at baseline, and at 1-, 3-, and 6-month follow-ups is presented in Table 1.

The most common feature at baseline was the presence of a small neovascular tuft within the outer retina slab, present in all 12 cases. Superjacent within the deep capillary plexus, dilated vessels were found in 7/12 (58%) eyes. In the choriocapillaris at the corresponding location, the lesion was identified in 9/12 (75%) eyes. Three of them displayed a perilesional hyporeflective halo. Examples of these clinical signs are presented in Figure 1.

Mean lesion area at baseline was 0.079 ± 0.072 mm^2^ in the outer retina slab, and 0.10 ± 0.16 mm^2^ in the choriocapillaris slab.

On the B-scans, Aouter retinal flow signal was detected on 9/12 (75%) eyes, RPE effraction was noted in 8/12 (67%) eyes, with anastomotic downward flow in 5/12 (42%) eyes. HRF were present at baseline in 6/12 (50%) eyes, three of which contained flow signal.

At 1 month after the first IVI of Brolucizumab, only one (8%) dilated vessel was found in the deep plexus (*p* = 0.031). Outer retinal vascular tufts were still present in 3 eyes (*p* < 0.01). Mean flow area was 0.087 ± 0.26 mm^2^ (*p* = 0.06). In the choriocapillaris, 5 eyes (42%) still had a visible lesion and mean lesion area was 0.17 ± 0.29 mm^2^ (*p* = 0.087). Only one perilesional hyporeflective halo subsisted (*p* = 0.86). On B-scans, 4/12 (33%) eyes still showed residual flow at the level of the outer retina (*p* = 0.125). RPE effraction was present in 4/12 (33%) eyes (*p* = 0.125), HRF were still present in two (17%) eyes (*p* = 0.5), and only one (8%) downward anastomotic flow was detected (*p* = 0.125).

At 3 month-visit, one (8%) dilated vessel was found in the deep capillary plexus (*p* = 0.031). 3/12 (25%) eyes still displayed outer retinal vascular tufts (*p* < 0.01). Mean flow area was 0.002 ± 0.0039 mm^2^ (*p* < 0.01). The lesion was still visible in one (8%) eye at the level of the choriocapillaris (*p* < 0.01). Mean area was 0.037 ± 0.13 (*p* = 0.086). Perilesional hyporeflective halo was not found in any eyes (*p* = 0.250). On B-scans, 3/12 (25%) eyes showed some flow at the level of the outer retina (*p* = 0.031). RPE effraction was still present in 5/12 (42%) eyes (*p* = 0.25). 3/12 eyes had HRF (*p* = 0.25), and downward anastomotic flow was detected in two (17%) eyes (*p* = 0.25).

At 6 months, 3/12 (25%) eyes showed a dilated vessel in the deep capillary plexus (*p* = 0.22). The lesion in the outer retina was found in 5/12 (42%) eyes (*p* = 0.016) and mean flow area was 0.03 ± 0.055 mm^2^ (*p* = 0.18). In the choriocapillaris, 5/12 (42%) eyes had a visible lesion (*p* = 0.29), with a mean flow area of 0.08 ± 0.13 mm^2^ (*p* = 0.59). On B-scans, outer retinal flow was found in 4/12 (33%) eyes (*p* = 0.18). 3/12 (25%) eyes had an RPE effraction (*p* = 0.125), 5/12 (42%) eyes had HRF (*p* = 1), and only one (8%) eye showed a downward flow (*p* = 0.125).

The mean CRT at baseline was 400 ± 64 µm, and decreased to 342 ± 53 µm, 329 ± 18 µm, and 289 ± 23 µm at 1-, 3-, and 6-month follow-up visits, respectively. Moreover, 5/12 (42%) eyes presented intraretinal fluid, 3/12 (25%) eyes presented subretinal fluid, 2/12 (17%) eyes showed both intra and subretinal fluid, and in 2/12 (17%) eyes there was also sub-RPE fluid. At 1 month follow-up 9/12 (75%) eyes showed a dry macula. At 3-month visit in 10/12 (83%) eyes no fluid was detected, and at 6 months 9/12 (75%) eyes presented absence of any fluid (Table 2).

## 4. Discussion

Type 3 MNVs correspond to an intraretinal neovascularization [2], arising from the retinal deep capillary plexus [5]. These lesions are usually small in size and exudative activity is minimal in the early stages of the disease [3,5,9]. These factors combined with the fact that the visualization of the deep capillary plexus is limited with fluorescein angiography [10] made the study of type 3 lesions difficult with traditional imaging modalities. The advent of OCT and, more recently OCTA, helped in the understanding of the disease origin and its natural evolution [5,9,11,12]. Indeed, OCTA is particularly suitable in this condition, as it allows detailed imaging of the deep capillary plexus and choriocapillaris. In pre-clinical stages, vascularized hyperreflective foci (HRF) located at the level of the deep capillary plexus can be observed [5]. These HRF are thought to correspond to an early intraretinal vasoproliferative process [5]. As the disease progresses, these HRF can migrate towards the RPE and the sub-RPE space to form an anastomosis (retino-retinal, or retino-choroidal) [5,13]. At this point, type 3 lesions can be recognized as bright tufts located at the level of the outer retinal layer on en face imaging, with tiny curvilinear vessels [9,12]. Within the choriocapillaris, a glomerule-like lesion can also be observed [13].

In this retrospective case series, we evaluated the short-term evolution of type 3 MNV on OCTA following Brolucizumab IVI. At baseline, we found the most consistent sign on OCTA was the presence of small vascular tufts within the outer retinal slab (100% of cases). A corresponding lesion was present in the subjacent choriocapillaris in 75% of cases, probably dependent on the evolution of the disease and the progression of the downward anastomosis. On B-scan, outer retinal flow signal was the most frequent sign, present in 75% of cases.

We found a good response to treatment with complete disappearance of detectable flow within the lesions at the level of the outer retina in 75% of cases at three months (*p* < 0.01). Although more lesions could be detected at 6 months, the final disappearance rate of 58% was still statistically significant (*p* = 0.016). In the choriocapillaris, 89% of the lesions detected at baseline disappeared at month 3 (*p* < 0.01). In a similar fashion to the outer retina, more lesions were visible in the choriocapillaris at month 6 and the eventual disappearance rate was not statistically significant. These findings suggest that a purely intraretinal early stage type 3 lesion could be more sensitive to anti-VEGF therapy, but once it has anastomosed within the choriocapillaris, it may become more resistant to treatment. In terms of lesion area, the difference was only statistically significant for the outer retinal lesion at 3 months (*p* < 0.01). This may be due to the inconsistent lesion sizes at baseline and during the follow-up. Indeed, as type 3 lesions are usually small in size [3,5,9], slight measurement incongruences can lead to high variabilities. Perilesional halo was also not frequent enough at baseline to draw conclusions on the significance of its disappearance. Finally, on the B-scans, the only parameter that yielded statistically significant reductions was the presence of an outer retinal flow at month 3 (*p* = 0.031), which is in accordance with the en face response.

As with any MNV type in nAMD, the cornerstone of treatment is IVI of anti-VEGF drugs. Before the advent of brolucizumab, studies with other anti-VEGF molecules have shown good efficacy on type 3 MNV appearance on OCTA [12,13,14]. Phasukkijwatana et al. [14] showed the complete regression of the lesion on OCTA in 5 of 17 cases (29%) in patients treated with aflibercept (2 cases), ranibizumab (2 cases), and bevacizumab (1 case). They noted that the presence of large feeder vessels was linked with resistance to treatment. In a study by Miere et al., 7 of 15 cases (47%) of patients treated with ranibizumab or aflibercept for type 3 MNV showed a complete disappearance of flow signal on OCTA at 12 months follow-up [13]. Mean number of injections was 5.9 following a *pro re nata* treatment regimen and interestingly, 40% of patients still had exudative activity at 12 months [13]. In Tan et al.’s study, however, only 1 out of 9 cases (11%) of treated eyes showed a complete resolution of the vascular tuft on en face OCTA, three weeks after one ranibizumab IVI (follow-up of the 9 cases ranged from 3 to 20 weeks after treatment) [12].

Recurrences of exudative activity and re-treatment decisions thereof are currently mostly based on OCT findings and to date, no consensus exists on an OCTA-guided treatment plan. Our findings show that disappearance of flow within the lesion on OCTA may lead to total absence of exudative recurrence, within the follow-up period of the study. This is also supported by other studies on OCTA [12,13,14]. However, it has also been shown that despite good initial response on OCTA scans, later recurrence was still possible [12], so careful follow-up should be warranted in absence of a proactive treatment plan. It is currently unclear whether the disappearance of signal on OCTA corresponds to the complete occlusion of the neovascular tuft, or a reduced lesion activity and flow speed, decelerating below the detection threshold. It could be assumed that in cases where no recurrences were noted, the neovascular complex was completely shut off, whereas in cases with later recurrences, the lesion’s activity decreased, without disappearing entirely. This outlines a potential dichotomy within the disease evolution: either the lesion resolves early, and treatment is no longer necessary, or long-term retreatment may be required. Current limits in OCTA technologies represent a barrier to reaching a definitive answer. This theory is in alignment with other studies [14,15] where it seems likely that reversal of the disease is possible during early stages of the neovascularization process, but as the lesion progresses and the anastomosis matures to form larger feeder vessels, regression becomes less common. Indeed, it has been shown that mature vessels in general do not regress with anti-VEGF therapy, probably owing to their adequate pericyte coverage providing a local supply of VEGF [16].

Brolucizumab is a novel anti-VEGF, which is composed by a single-chain antibody fragment. This conformation allows for a retained binding capacity, with light supporting structural elements needed [17]. This reflects on its molecular weight of 26 kDa, about 1.8× lighter than ranibizumab and about 4× lighter than aflibercept [17]. As a consequence, a much higher concentration of anti-VEGF can be achieved within the same dose of injection. Additionally, the small molecular size allows for a good penetration within the retina and choroid and at a fast rate [17]. Furthermore, brolucizumab exhibits a good binding affinity to VEGF-A, allowing it to reach the same potency as ranibizumab at a much lower concentration [17]. All these elements may explain its presumably greater anti-VEGF potency and longer durability. The mean number of injections in our cohort might reflect the properties of this molecule.

In this study, we found a high rate of lesion regression after treatment. While this can happen with other anti-VEGF [12,13,14], the higher rate of regression obtained with brolucizumab could be attributable to the high potency of the drug. Brolucizumab may allow for complete VEGF and neovascular activity inhibition and therefore stop the progression of the lesion towards a stage where reversal is no longer possible.

We acknowledge that 12 included eyes is a small number to draw definitive conclusions. However, this is a pilot study and it may guide future research with greater numbers. Moreover, six months is a too short time and a longer follow-up is needed to determine the durability of this treatment for type 3 MNV. Besides, inter- and intra-observer variability in detection and measurement of the vascular network on OCTA scans can be high and an automated method for detecting the new vessels would be warranted. On the other hand, despite these limitations, the quality of the imaging and the parameters considered allow us to have solid short-term results and represent a consistent and reproducible method for a long-term and larger study. Moreover, limited data are at the moment available about this subtype of MNV and brolucizumab therapy.

## 5. Conclusions

In conclusion, we have shown a good short-term efficacy of brolucizumab IVI on OCTA in patients with early stage type 3 MNV in the context of nAMD. These findings support the idea that type 3 lesions seem to display an early sensitivity to anti-VEGF and are susceptible to occlusions. As brolucizumab displays great potency, it may be responsible for a higher proportion of lesion regressions. Further studies are needed to validate the long-term efficacy of this approach.

## Figures and Tables

**Figure 1 medicina-58-01180-f001:**
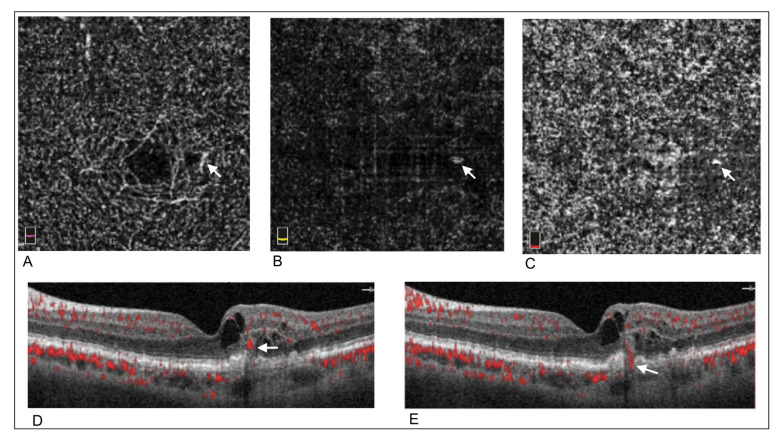
Example of optical coherence tomography angiography (OCTA) signs at baseline in a patient with a type 3 lesion. White arrow shows the lesion on each corresponding scans. (**A**) Dilated vessels can be observed in the retinal deep vascular plexus. (**B**,**C**) A small vascular tuft is present in the outer retinal slab and choriocapillaris slabs, respectively. (**D**) B-scan showing the presence of flow signal within the outer retina. (**E**) B-scan adjacent to D showing an anastomotic flow passing from the retina through an opened retinal pigment epithelium.

**Table 1 medicina-58-01180-t001:** Signs on optical coherence tomography angiography (OCTA) at baseline, at one month, three months, and six months after Brolucizumab treatment.

OCTA Sign			Baseline	1 Month	3 Months	6 Months
En face	Deep plexus	Dilated vessel	7	1 (*p* = 0.031)	1 (*p* = 0.031)	3 (*p* = 0.22)
	Outer retina	Small tuft	12	3 (*p* < 0.01)	3 (*p* < 0.01)	5 (*p* = 0.016)
		Area (mm^2^)	0.079	0.087 (*p* = 0.06)	0.002 (*p* < 0.01)	0.03 (*p* = 0.18)
	Choriocapillaris	Small tuft	9	5 (*p* = 0.125)	1 (*p* < 0.01)	5 (*p* = 0.29)
		Area (mm^2^)	0.10	0.17 (*p* = 0.86)	0.037 (*p* = 0.086)	0.08 (*p* = 0.59)
		Perilesional hyporeflective halo	3	1 (*p* = 0.625)	0 (*p* = 0.250)	2 (*p* = 1)
B-scan		Outer retina flow	9	4 (*p* = 0.125)	3 (*p* = 0.031)	4 (*p* = 0.18)
		RPE effraction and/or kissing sign	8	4 (*p* = 0.125)	5 (*p* = 0.25)	3 (*p* = 0.125)
		HRF	6	2 (*p* = 0.5)	3 (*p* = 0.25)	5 (*p* = 1)
		Downward flow	5	1 (*p* = 0.125)	2 (*p* = 0.25)	1 (*p* = 0.125)

*p*-values correspond to comparison from baseline. Wilcoxon signed rank test was used for continuous variables and McNemar’s test for categorical values. OCTA = Optical coherence tomography angiography, RPE = retinal pigment epithelium, HRF = hyperreflective foci.

**Table 2 medicina-58-01180-t002:** Characteristics on optical coherence tomography (OCT) B-scan at baseline, at one month, three months, and six months after Brolucizumab treatment.

OCT B-Scan	Baseline	1 Month	3 Months	6 Months
Mean CRT ± SD (microns)	400 ± 64	342 ± 53	329 ± 18	289 ± 23
Dry macula, *n* (%)	0	9 (75%)	10 (83%)	9 (75%)
IRF, *n* (%)	5 (42%)	0	1 (8%)	1 (8%)
SRF, *n* (%)	3 25%)	2 (17%)	1 (8%)	1 (8%)
IRF + SRF, *n* (%)	2 (17%)	0	0	0
IRF + SRF + PED, *n* (%)	2 (17%)	1 (8%)	1 (8%)	1 (8%)

OCT = Optical coherence tomography, CRT = central retinal thickness, IRF = intraretinal fluid, SRF = subretinal fluid, PED = pigment epithelium detachment.

## Data Availability

The datasets used and/or analyzed during the current study are available from the corresponding author on reasonable request.

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
