# Peer review of "Early OCTA Changes of Type 3 Macular Neovascularization Following Brolucizumab Intravitreal Injections"

_medicina, 2022, doi:10.3390/medicina58091180_

Round 1

Reviewer 1 Report

1. The authors provide a description of OCTA changes following brolucizumab in type 3 MNV. However, as mentioned by the authors, the sample size and follow-up durations are significantly less to come to a definitive conclusion. In addition to this, considering the fact that the inter and intra-observer variability in detection and measurement of the small vascular networks of type 3 MNV can be quite high, the study fails to provide any concrete evidence regarding the early effect of brolucizumab.

Nonetheless, the topic is highly relevant and I appreciate the authors for their work.

2. Although the aim of the study is to analyze the OCTA characteristics, it would be useful for the readers to understand how the macular thickness changed with time. Also, how many eyes had dry macula at each time point.

3. It was surprising to see an increase in network characteristics at 6 months, considering the fact that change in flow signal was one of the retreatment criteria. Weren't these eyes treated when the network size was seen to increase? Page 2 line 69: ".../or a new flow signal was evident at the level of the outer retina or choriocapillaris on 69 OCTA."It would be important to describe the follow-up pattern and the number of injections in this subgroup where the network characteristics increased.

Author Response

Comment 1: The authors provide a description of OCTA changes following brolucizumab in type 3 MNV. However, as mentioned by the authors, the sample size and follow-up durations are significantly less to come to a definitive conclusion.

Answer 1: We are perfectly aware that the sample size and the follow-up are too small to come to a definitive conclusion. We stated this in the limitations paragraph (page 7, line 284).

Comment 2: In addition to this, considering the fact that the inter and intra-observer variability in detection and measurement of the small vascular networks of type 3 MNV can be quite high, the study fails to provide any concrete evidence regarding the early effect of brolucizumab.

Nonetheless, the topic is highly relevant and I appreciate the authors for their work.

Answer 2: We recognize that there is no a standardized method for measurement of small vascular networks on OCTA images. As reported in the methods section (page 2, line 86), two readers analyzed the images and in case of discrepancies a third retina specialist solved them. We add a sentence in the discussion of limitations (page 7, line 287) as follows: “Besides, inter- and intra-observer variability in detection and measurement of the vascular network on OCTA scans can be high and an automated method detecting the new vessels would be warranted.”

Comment 3: Although the aim of the study is to analyze the OCTA characteristics, it would be useful for the readers to understand how the macular thickness changed with time. Also, how many eyes had dry macula at each time point

Answer 3: We added the mean central retinal thickness changes (CRT) and the number of the eyes with dry macula at each time point. In particular, we updated the methods section (page 3, line 107) with the following sentence: “Qualitative analysis of the B-scan OCT was also performed in terms of presence/absence of intra or subretinal fluid or pigment epithelium detachment. The central retinal thickness (CRT) was measured in the 1 mm central ETDRS grid using the inbuilt automatic tool.”

In the results section (page 5, line 181) we added the following sentence: “The mean CRT at baseline was 400 ± 64 µm, and decreased to 342 ± 53 µm, 329 ± 18 µm, and 289 ± 23 µm at 1-, 3--, and 6-month follow-up visits, respectively. Moreover, 5/12 (42%) eyes presented intraretinal fluid, 3/12 (25%) eyes presented sub-retinal fluid, 2/12 (17%) eyes showed both intra and subretinal fluid, and in 2/12 (17%) eyes there was also sub-RPE fluid. At 1 month follow-up 9/12 (75%) eyes showed a dry macula. At 3-month visit in 10/12 (83%) eyes no fluid was detected, and at 6 months 9/12 (75%) eyes presented absence of any fluid. (Table 2)”. Moreover, Table 2 summarizes the OCT B-scan characteristics (page 5).

Comment 4:  It was surprising to see an increase in network characteristics at 6 months, considering the fact that change in flow signal was one of the retreatment criteria. Weren't these eyes treated when the network size was seen to increase? Page 2 line 69: ".../or a new flow signal was evident at the level of the outer retina or choriocapillaris on 69 OCTA."It would be important to describe the follow-up pattern and the number of injections in this subgroup where the network characteristics increased.

Answer 4: We thank the reviewer for this comment. In the follow-up the eyes were retreated when recurrent intra- or sub-retinal fluid was present on OCT B-scan with or without new flow sign on OCTA, while the evidence of a change in the flow signal on OCTA alone was not a retreatment criteria. We clarified this point on the methods section (page 2, line 72) as follow: “Flow signal changes on OCTA alone without recurrence of fluid on OCT B-scan was not a retreatment criteria.”

In the result section (page 3, line 125), we reported the number of injections and we added the following sentence: “Only three out of 12 eyes received one IVI in the maintenance phase after the first three monthly treatments. All three eyes presented new flow signal on outer retina or choriocapillaris and recurrent intra or subretinal fluid.”

Comment 5: Moderate English changes required.

Answer 5: English revision was performed by a professional reviewer.

Reviewer 2 Report

Authors reported initial results of brolucizumab IVI on type 3 neovascularization in nAMD, evaluated by OCTA. They showed that brolucizumab IVI shows a good short-term efficacy for the treatment of type 3 neovascularizations. Their data are interesting and may have impacts for retina specialists. I have following comments:

Comment 1:

Authors should provide more information on the cases – gender, presence or absence of diabetes mellitus, hypertension, and hyperlipidemia.

Comment 2:

Did this study include a history of IOL or anti-VEGF therapy?

Comment 3:

Was there any systemic or ocular (e.g., intraocular inflammation) side effects after brolucizumab IVI?

Author Response

Authors reported initial results of brolucizumab IVI on type 3 neovascularization in nAMD, evaluated by OCTA. They showed that brolucizumab IVI shows a good short-term efficacy for the treatment of type 3 neovascularizations. Their data are interesting and may have impacts for retina specialists. I have following comments:

Comment 1: Authors should provide more information on the cases – gender, presence or absence of diabetes mellitus, hypertension, and hyperlipidemia.

Answer 1: We added the suggested information on the results section (page3, line 119) “Six (50%) patient were male. One (8%) subject had diabetes type 2 for 10 years, 5 (42%) and 3 (25%) patients received treatment for blood hypertension and hypercholesterolemia, respectively. Six (50%) eyes were pseudophakic with cataract surgery performed more than 5 years before.”

Comment 2: Did this study include a history of IOL or anti-VEGF therapy?

Answer 2: As mentioned on the previous answer, we added information regarding IOL and cataract surgery (page 3, line 121).

Previous anti-VEGF treatment was an exclusion criteria. Therefore there was no history of anti-VEGF therapy, as already stated in the methods section (page 2, line 58) “…of patients with newly diagnosed nAMD and type 3 MNV…” Only naïve eyes were included.

Comment 3: Was there any systemic or ocular (e.g., intraocular inflammation) side effects after brolucizumab IVI?

Answer 3: In the results section (page 3, line 123), we already reported the absence of both systemic and intraocular adverse event during the 6 months follow-up:No systemic or intraocular adverse events were documented throughout the follow-up period.”

Round 2

Reviewer 1 Report

The queries have been well addressed. There are no further comments.